# N-Ary Quantization for CNN Model Compression and Inference Acceleration

## Abstract

The tremendous memory and computational complexity of Convolutional Neural Networks (CNNs) prevents the inference deployment on resource-constrained systems. As a result, recent research focused on CNN optimization techniques, in particular quantization, which allows weights and activations of layers to be represented with just a few bits while achieving impressive prediction performance. However, aggressive quantization techniques still fail to achieve full-precision prediction performance on state-of-the-art CNN architectures on large-scale classification tasks. In this work we propose a method for weight and activation quantization that is scalable in terms of quantization levels (n-ary representations) and easy to compute while maintaining the performance close to full-precision CNNs. Our weight quantization scheme is based on trainable scaling factors and a nested-means clustering strategy which is robust to weight updates and therefore exhibits good convergence properties. The flexibility of nested-means clustering enables exploration of various n-ary weight representations with the potential of high parameter compression. For activations, we propose a linear quantization strategy that takes the statistical properties of batch normalization into account. We demonstrate the effectiveness of our approach using state-of-the-art models on ImageNet.

## 1 Introduction

The increasing computational complexity and memory requirements of Convolutional Neural Networks (CNNs) have motivated recent research efforts in efficient representation and processing of CNNs. Several optimization and inference approaches have been proposed with the objective of model compression and inference acceleration. The primary aim of model compression is to enable on-device storage (e.g., mobile phones and other resource-constrained devices) and to leverage on-chip memory in order to reduce energy consumption (Horowitz, 2014), latency and bandwidth of parameter accesses. Inference acceleration can be achieved by lowering the precision of computations in terms of resolution, removing connections within networks (pruning), and specialized software/hardware architectures.

Quantized Neural Networks are optimization techniques where weights and/or activations of a neural network are transformed from 32-bit floating point into a lower resolution; for aggressive quantization techniques down to binary (Courbariaux et al., 2015; Rastegari et al., 2016) or ternary (Li & Liu, 2016; Zhu et al., 2016) representations. Although prior quantization techniques achieve full-precision accuracy on highly over-parameterized architectures (e.g., AlexNet, VGG) or toy tasks (e.g., MNIST, SVHN, CIFAR-10), there is still an unacceptable gap between extremely low bit-width representations and state-of-the-art architectures for real-world tasks. Furthermore, aggressive quantization approaches are usually designed for specific representations (e.g., binary or ternary), and are not scalable in the sense that they do not allow for more quantization levels (i.e., weight values) if required. Thus, accuracy degradation cannot be compensated without changes in the baseline architecture which includes deepening or widening the neural network, and/or using full-precision weights in the input and output layers, respectively.

In this work, we address the issue of accuracy degradation by introducing a scalable non-uniform quantization method for weights that is based on trainable scaling factors in combination with a

nested-means clustering approach. In particular, nested-means splits the weight distribution iteratively into several quantization intervals until a pre-defined discretization level is reached. Subsequently, all weights within a certain quantization interval are assigned the same weight (i.e., the scaling factor). Nested-means clustering tends to assign small weights to larger quantization intervals while less frequent larger weights are assigned to smaller quantization intervals. This improves classification performance which is in line with recent observations that larger weights carry more information than smaller weights (Han et al., 2015a). We evaluate our approach on state-of-the-art CNN architectures in terms of computational requirements and prediction accuracy using the ImageNet classification task.

The paper is structured as follows. In Sec. 2 and Sec. 3 related work and background on inference acceleration is discussed. Weight and activation quantization is presented in Sec. 4 and Sec. 5, respectively. Experimental results for ImageNet are shown in Sec. 6. Sec. 7 concludes the paper.

## 2 RELATED WORK

CNN quantization is an active research area with various approaches and objectives. In this section, we briefly review the most promising strategies from different categories. We distinguish between two orthogonal qualitative dimensions: (i) Approaches that only quantize the weights and approaches that quantize both the weights and the activations, and (ii) scalable approaches that allow for different bit-widths and non-scalable approaches that are designed for specific weight representations.

**Non-scalable weight-only quantization**: Courbariaux et al. (2015) introduced Binary Connect (BC) in which they constrain weights to either -1 or 1. Rastegari et al. (2016) proposed Binary Weight Networks (BWNs), which improves BC by introducing channel-wise scaling factors. Li & Liu (2016) introduced Ternary Weight Networks (TWNs), in order to account for the accuracy degradation of BWNs. Zhu et al. (2016) proposed Trained Ternary Quantization (TTQ), extending TWNs by non-uniform and trainable scaling factors.

**Scalable weight-only quantization**: Han et al. (2015a) proposed Deep Compression, a method that leverages pruning, weight sharing and Huffman Coding for model compression. Zhou et al. (2017) proposed Incremental Network Quantization (INQ) to quantize pre-trained full-precision DNNs to zeros and powers of two. This is accomplished by iteratively partitioning the weights into two sets, one of which is quantized while the other is retrained to compensate for accuracy degradation.

**Non-scalable quantization for weights and activations**: Binarized Neural Networks (Courbariaux & Bengio, 2016), XNOR-Net (Rastegari et al., 2016), Bi-Real Net (Liu et al., 2018b) and ABC-Net (Lin et al., 2017) quantize both weights and activations to either -1 or 1. Cai et al. (2017) proposed low-precision activations using Half-Wave Gaussian Quantization (HWGQ), and relies on BWN for weight binarization. Based on symmetric binary and ternary weights as proposed by Zhu et al. (2016), Faraone et al. (2018) introduced Symmetric Quantization (SYQ) by pixel-wise scaling factors and fixed-point activation quantization.

**Scalable quantization for weights and activations**: DoReFa-Net (Zhou et al., 2016), PACT (Choi et al., 2018b) and PACT-SAWB (Choi et al., 2018a) allows weights and activation to be variable configurable. Baskin et al. (2018) proposed uniform noise injection for non-uniform quantization (UNIQ) of both weights and activations. Lin et al. (2017) use multi-bit quantization. Zhang et al. (2018) proposed an adaptively learnable quantizer (LQ-Nets) and achieves state-of-the-art accuracy as of today.

In this work, we focus on a scalable quantization of weights and activations, with the main objective of maintaining accuracy while exploiting model compression and reduced computational complexity for inference acceleration. Our approach differs from prior work as we introduce a novel quantization strategy that is easy to compute, maintains state-of-the-art prediction accuracy, and has significantly less impact on training time. In particular, our approach enables various non-uniform n-ary weight representations with a high amount of sparsity.

## 3 BACKGROUND ON QNN INFERENCE

A quantization function maps the filter weights and/or the inputs to the convolution within a quantization interval to a quantization level $\alpha_i$. A quantizer is uniform if $\alpha_{i+1} - \alpha_i = c, \forall i$, where $c$ is a constant quantization step. If both filter weights and input data are quantized uniformely using the same number of quantization levels, then the scalar products in the QNN forward convolution can be realized by reduced-precision integer/fixed-point (Jouppi et al., 2017) or half-precision floating-point (Markidis et al., 2018)) operations.

Recent research (Cai et al., 2017) shows, however, that input data requires more quantization levels than the filter weights in order to achieve single-precision floating point accuracy. Few-bit integer scalar products with different amounts of quantization levels can be implemented using the bit-serial/bit-plane approach (Courbariaux & Bengio, 2016) which is done using bitwise XNOR or AND followed by popcount operations.

Non-uniform quantization usually achieves better classification accuracy than uniform quantization. Furthermore, a certain amount of sparsity (percentage of zero elements) in the filter weights potentially reduces the number of operations without affecting the prediction performance (Han et al., 2015a). In order to account for these representations, Han et al. (2016) proposes the Efficient Inference Engine, a specialized hardware CNN inference accelerator that leverages non-uniform weights, uniform activations, and sparsity.

Another approach are reduce-and-scale architectures that can be efficiently realized in software on general-purpose hardware (Schindler et al., 2018). The basic idea of reduce-and-scale inference is to sum all equally weighted inputs and to multiply the result with the respective weight. This approach efficiently leverages sparsity in weights and requires only one multiplication per quantization level and per output feature, shifting the computational bottleneck to additions. Furthermore, weight quantization can be non-uniform, and quantization levels can be tailored for weights and activations, respectively. Here, we focus on accelerating the inference based on the reduce-and-scale approach.

## 4 WEIGHT QUANTIZATION

For weight quantization, we employ the common strategy of maintaining full-precision weights for training that are quantized during forward propagation. The gradient of the full-precision weights is approximated by backpropagating through quantization functions using the straight-through gradient estimator (STE) (Bengio et al., 2013). The gradient is subsequently used to update the full-precision weights. For inference, the full-precision weights are discarded and only the quantized weights are used for model compression and inference acceleration. In this section we describe the quantization strategy that is used at forward propagation.

### 4.1 THRESHOLDING AND SCALING FACTORS

The classification accuracy of aggressive quantization techniques heavily relies on the usage of scaling factors as model capacity is improved significantly. For binary weights, Rastegari et al. (2016) propose one uniform scaling factor $\alpha_k^l$ per layer $l$ and output feature map $k$, which is calculated as the mean of absolute floating-point weights, i.e., $\alpha_k^l = (\sum_{w \in \mathbf{W}_k^l} |w|)/|\mathbf{W}_k^l|$ where $\mathbf{W}_k^l$ denotes the set of weights connected to output feature map $k$. For ternary weights, Zhu et al. (2016) propose two trainable non-uniform scaling factors $\alpha_+^l$ and $\alpha_-^l$ per layer that are determined by gradient descent. This adjusts the scaling factors so as to minimize the given loss function while at the same time increasing the model capacity due to non-uniform scaling. Therefore, we adopt trainable scaling factors in our method. Let $\delta_{-K_n}^l < \ldots < \delta_{-1}^l < 0 < \delta_{+1}^l < \ldots < \delta_{+K_p}^l$ be a set of interval thresholds that partition the real numbers into intervals

$$\mathbf{\Delta}_{-K_n}^l = \left(-\infty, -\delta_{-K_n}^l\right), \ \mathbf{\Delta}_{-i}^l = \left[\delta_{-i-1}^l, \delta_{-i}^l\right), \ \mathbf{\Delta}_{+i}^l = \left[\delta_{+i}^l, \delta_{+i+1}^l\right), \ \mathbf{\Delta}_{+K_p}^l = \left[\delta_{+K_p}^l, \infty\right).$$
(1)

If the zero weight should be explicitly modeled, we define an interval $\mathbf{\Delta}_0^l = [\delta_{-1}^l, \delta_{+1}^l)$. If no explicit zero weight is modeled, we define two intervals $\mathbf{\Delta}_{-0}^l = [\delta_{-1}^l, 0)$ and $\mathbf{\Delta}_{+0}^l = [0, \delta_{+1}^l)$. To each interval $\mathbf{\Delta}_i^l$, we assign a trainable scaling factor $\alpha_i^l$ that is used to quantize the weights as

$w^q = \alpha_i^l \Leftrightarrow w \in \mathbf{\Delta}_i^l$. During training, we update the scaling factors $\alpha_i^l$ using gradients computed as

$$\frac{\partial E}{\partial \alpha_i^l} = \sum_{w^l \in \mathbf{\Delta}_i^l} \frac{\partial E}{\partial (w^l)^q}, \tag{2}$$

where $E$ denotes the loss function and $(w^l)^q$ denotes the quantized weights. In case the zero weight is modeled, we have a fixed scaling factor $\alpha_0^l = 0$ that is not updated during gradient descent. Finding good interval thresholds $\delta$ is essential for prediction performance and will be discussed in Sec. 4.3.

## 4.2 NON-UNIFORM WEIGHT REPRESENTATIONS

Allowing the weights to be non-uniform enables various explorations of weight representations. Since weight distributions tend to be symmetric around zero (see Fig. 1), good quantized weight representations also exhibit a symmetry around zero. Candidates for such representations are summarized in Table 1.

Table 1: Different non-uniform n-ary weight representations

| Notation | Representation | Bit width (dense) | Bit width (sparse) |
|---|---|---|---|
| Binary | $\{\alpha_{-0}, \alpha_{+0}\}$ | 1 | – |
| Ternary | $\{\alpha_{-1}, 0, \alpha_{+1}\}$ | 2 | 1 |
| Quaternary | $\{\alpha_{-1}, \alpha_{-0}, \alpha_{+0}, \alpha_{+1}\}$ | 2 | – |
| Quaternary+ | $\{\alpha_{-1}, 0, \alpha_{+1}, \alpha_{+2}\}$ | 2 | 2 |
| Quaternary- | $\{\alpha_{-2}, \alpha_{-1}, 0, \alpha_{+1}\}$ | 2 | 2 |
| Quinary | $\{\alpha_{-2}, \alpha_{-1}, 0, \alpha_{+1}, \alpha_{+2}\}$ | 3 | 2 |

Binary and ternary representations gained a lot of interest lately due to their high compression ratios and, in view of their low expressiveness, relatively good prediction performance. On large-scale classification tasks, however, only ternary weights demonstrate a prediction accuracy similar to full-precision weights. Furthermore, for ternary representations weight pruning is possible with little impact on prediction accuracy which additionally reduces the computational complexity of inference. In this work, we also explore other n-ary weight representations that facilitate compression levels similar to ternary weights while significantly improving model capacity. For instance, quaternary weights can also be encoded with only two bits, but introduce either an additional positive (quaternary+) or negative (quaternary-) scaling factor, respectively. Quinary weights extend ternary weights by one positive and one negative value, but are still encoded with only two bits in a sparse format. In a sparse format, only the indices of non-zero weight entries are stored such that two bits are sufficient to represent the non-zero values.

## 4.3 NESTED-MEANS CLUSTERING

For an optimal approximation, weight clustering is required to partition a set of weights that are later represented by a single discrete value per cluster (cf. Sec. 4.1). The clustering can be implemented either statically (once before training) or dynamically (repeatedly during training) by calculating thresholds $\delta$ which represent the boundaries of the respective cluster.

The static approach has the advantage of allowing iterative clustering algorithms to be applied (e.g., k-means clustering or Lloyd's algorithm (Lloyd, 1982)) that are able to find an optimal solution for the cluster assignment. However, as quantization lowers the resolution and therefore changes in the weight distribution, quantization requires re-training to compensate for this loss of information. As a consequence, the optimal solution found by an iterative algorithm will become non-optimal during the following re-training process. Lowering the learning rate for re-training can diminish heavy changes in the weight distribution, at the cost of longer time to converge and the risk to get stuck at plateau regions, which is especially critical for trainable scaling factors (Eq. 2). Applying an iterative clustering approach repeatedly during training is practically infeasible, since it causes a dramatic increase in training time.

A practical useful clustering solution is to calculate cluster thresholds during training based on the maximum absolute value of the full-precision weights per layer (Zhu et al., 2016): $\delta_l = t \cdot \max(|w_l|)$, where $t$ is a hyperparameter. This approach is beneficial because it defines cluster thresholds which are influenced by large weights that were shown to play a more important role than smaller weights (Han et al., 2015b). Furthermore, training time is virtually unaffected by this rather simple calculation. However, having an additional hyperparameter $t_i$ for each scaling factor $\alpha_i$ renders the mandatory hyperparameter tuning infeasible. Furthermore, the sensitivity to the maximum value results in aggressive threshold changes caused by weight updates, possibly even preventing the network from converging.

In order to overcome these issues, we propose a symmetric nested-means clustering algorithm for assigning full-precision weights to a set of quantization clusters. Since weight distributions are typically symmetric around zero, we divide the weights into a positive and a negative cluster ($\mathbf{I}^l_{+1}$ and $\mathbf{I}^l_{-1}$). These clusters are then divided at their arithmetic means $\delta^l_{+i}$ and $\delta^l_{-i}$ into two subclusters – for each cluster we obtain an inner cluster and an outer cluster containing the tail of the distribution. The subclusters containing the tail of the distribution ($\mathbf{I}^l_{+i+1}$ and $\mathbf{I}^l_{-i-1}$) are repeatedly divided at their arithmetic means until the targeted number of quantization intervals is reached. More formally, nested-means clustering iteratively computes

$$\delta^l_{+i} = \frac{1}{|\mathbf{I}^l_{+i}|} \sum_{j \in \mathbf{I}^l_{+i}} w^l_j \quad \text{and} \quad \delta^l_{-i} = \frac{1}{|\mathbf{I}^l_{-i}|} \sum_{j \in \mathbf{I}^l_{-i}} w^l_j \tag{3}$$

$$\mathbf{I}^l_{+i+1} = \{j | w^l_j \geq \delta^l_{+i}\} \quad \text{and} \quad \mathbf{I}^l_{-i-1} = \{j | w^l_j < \delta^l_{-i}\}, \tag{4}$$

starting with $\mathbf{I}^l_{+1} = \{j | w^l_j \geq 0\}$ and $\mathbf{I}^l_{-1} = \{j | w^l_j < 0\}$. The whole clustering process is shown in Fig. 1 on the example of seven quantization clusters.

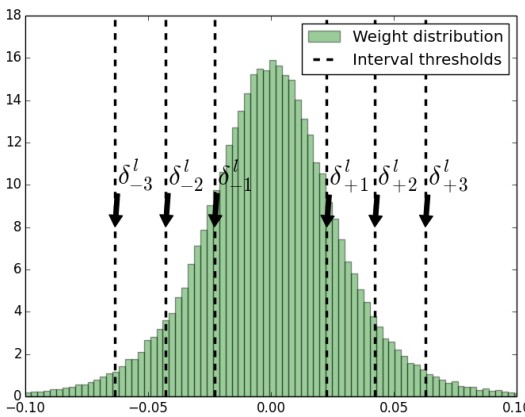

Figure 1: Nested-means intervals of a trained ResNet layer.

Nested-means clustering naturally defines the cluster thresholds in a way that the cluster intervals become smaller for larger weights. Although this might seem counter-intuitive since most weights are close to zero and the sum of quantization errors could be made smaller by using a higher quantization resolution around zero, this is actually beneficial as large weights were shown to play a more important role than smaller weights (Han et al., 2015a). At the same time the mean is less sensitive to weight updates than the maximum absolute weight value (Zhu et al., 2016), allowing for better convergence. Last, nested-means clustering is hyperparameter-free and requires only one arithmetic mean per cluster, which is computationally efficient.

## 5 ACTIVATION CLIPPING

The computational heavy lifting of the reduce-and-scale inference is reducing equally weighted inputs to a convolution layer (as discussed in Sec. 3). Lowering the bit width of the inputs enables

better usage of data level parallelism (performing the same operation on multiple inputs simultaneously) and results in less operations and memory accesses. We use a linear quantization scheme for activations that allows the required low bit-width additions to be performed using integer arithmetic on commodity devices.

The rectified linear unit (ReLU) activation function $f(x) = \max(0, x)$ is a commonly used non-linear activation function due its computational efficiency and its ability to alleviate the vanishing gradient problem. However, the ReLU function produces unbounded outputs which potentially require a high dynamic range and are therefore difficult to quantize. A common solution for this is to clip the outputs in the interval $(0, \gamma]$:

$$Q_c(x) = \begin{cases} 0 : x \leq 0 \\ x : 0 < x \leq \gamma \\ \gamma : \gamma < x \end{cases} \quad . \tag{5}$$

When selecting the clipping parameter $\gamma$, a trade-off needs to be made: On the one side, small values of $\gamma$ produce gradient mismatches due to different forward and backward approximations in the clipped interval $(\gamma, \infty)$. On the other side, large values of $\gamma$ result in a large interval $(0, \gamma]$ that needs to be quantized. This is problematic if only a few bits are used as quantization errors might become large.

In order to define an appropriate clipping interval, we use the observation that pre-activations tend to have a Gaussian distribution (Ioffe & Szegedy, 2015). In a Gaussian distribution, most values lie within a rather small range and there are only a few outliers that yield a high absolute range. For instance, 99.7% of the values lie within three standard deviations $\sigma$ of the mean $\mu$ (Smirnov & Dunin-Barkovskiĭ, 1963). We find this empirical rule to be a good approximation to filter out outliers and define the clipping interval as $\gamma = \mu + 3\sigma$.

This approach approximates the ReLU function well but suffers from the drawback that $\mu$ and $\sigma$ need to be repeatedly calculated during training. In recent years, batch normalization became a standard tool to accelerate convergence of state-of-the-art CNNs (Ioffe & Szegedy, 2015). Batch normalization transforms individual pre-activations to approximately have zero mean and unit variance across all data samples. Cai et al. (2017) experimentally showed that the pre-activation distribution after batch normalization are all close to a Gaussian with zero mean and unit variance. Therefore, we propose to select a fixed clipping parameter $\gamma = 3$ as it results in a small quantization interval $(0, \gamma]$ while also keeping the number of clipped activations $x > \gamma$ small.

## 6 EXPERIMENTS

We applied the proposed quantization approach on ResNet (He et al., 2015) and a variant of the Inception network (Ioffe & Szegedy, 2015), trained on the ImageNet classification task (Russakovsky et al., 2015). We use TensorFlow (Abadi et al., 2016) and the Tensorpack library (Wu et al., 2016). For the ResNet network, the learning rate starts from 0.1 and is divided by 10 each $30 \times 100$ iterations, a weight decay of 0.0001, and a momentum of 0.9 is used. For the Inception network, we schedule learning rates following the configuration of Wu et al. (2016), a weight decay of 0.0001, and a momentum of 0.9. We use eight GPUs for training and a batch size of 64 per GPU. The quantized networks leverage initialization with pre-trained full-precision parameters. We quantize all convolutional layers and fully-connected layers except the input and output layers, respectively, to avoid accuracy degradation (Zhou et al., 2016; Zhu et al., 2016; Faraone et al., 2018; Zhang et al., 2018). A detailed comparison to reported results of various related methods is summarized in Appendix B.

### 6.1 WEIGHT QUANTIZATION

We evaluate our nested-means weight quantization on ternary, quaternary- and quinary representations (Table 1) as these representations have good compression/acceleration potential while achieving the best accuracy. Table 2 reports the validation accuracy of ResNet-18 and Inception-BN on the ImageNet task.

The training time increases with increasing quantization levels because of the additional computations of interval thresholds (Eq. 3). While the impact on training time is negligible for the Inception-

Table 2: Validation accuracy (Top1,Top5), increase in training time and sparsity (fraction of zero weights) for different weight representation of ResNet18 and Inception-BN on ImageNet.

|  | Weights | Activations | Training | Top1 | Top5 | Sparsity |
|---|---|---|---|---|---|---|
| ResNet-18 | bits | bits |  | % | % | % |
| Baseline | 32 | 32 | 1.0x | 70.4 | 89.5 | – |
| Quinary | 3 | 32 | 2.0x | 69.7 | 89.0 | 53 |
| Quaternary- | 2 | 32 | 1.6x | 69.2 | 88.7 | 57 |
| Ternary | 2 | 32 | 1.2x | 68.2 | 88.0 | 61 |
| Inception-BN | bits | bits |  | % | % | % |
| Baseline | 32 | 32 | 1.0x | 73.1 | 91.4 | – |
| Quinary | 3 | 32 | 1.1x | 71.7 | 90.3 | 54 |
| Quaternary- | 2 | 32 | 1.1x | 70.8 | 89.8 | 52 |
| Ternary | 2 | 32 | 1.0x | 69.6 | 89.1 | 58 |

BN model, the ResNet model shows an increase of up to 2.0x. This is caused by a better GPU utilization of ResNet which makes the overall training time more sensitive to additional operations.

## 6.2  ACTIVATION QUANTIZATION

In this section, we evaluate the activation quantization using ResNet-18 on ImageNet. We use the clipped ReLU (Eq. 5) and set $\gamma = 3$ (as discussed in Sec. 5) for quantized activations, and we use the ReLU without clipping and quantization for 32-bit activations. Table 3 reports the validation accuracy and the increase in training time for several activation bit-widths. The training time is only

Table 3: Validation accuracy (Top1,Top5) and increase in training time for different activation bit-width of ResNet-18 on ImageNet.

| Weights | Activations | Training | Top1 | Top5 |
|---|---|---|---|---|
| Ternary | 32 | 1.2x | 68.2 | 88.0 |
| Ternary | 8 | 1.2x | 68.1 | 88.0 |
| Ternary | 4 | 1.2x | 68.1 | 88.1 |
| Ternary | 2 | 1.2x | 66.2 | 86.7 |

influenced by the weight quantization, whereas the influence of activation quantization is neglibible. The efficiency of the activation quantization is also shown in Fig. 2. As can be seen, the learning curves of ternary weights and 32-bit and 4-bit activations are roughly identical which highlights the robustness of the proposed quantization approach. Only 2-bit activations result in a slight accuracy degradation, but also this case shows a stable learning behavior.

## 6.3  ABLATION STUDY

In order to show the effectiveness of the nested-mean clustering, we experimentally show the impact of the key components. First, we show the importance of the threshold robustness to weight changes and the ability to represent general n-ary weights with a comparison to Zhu et al. (2016). Then we reason about the effectiveness of nested-means clustering by comparing it to several combinations of a quantile clustering approach that is described below.

**Impact of threshold robustness and configurable quantization levels:**   As described in Section 4.3, defining cluster thresholds that are robust against updates in the underlying weight distribution is vital for convergence and the prediction performance. The approach of Zhu et al. (2016) defines the thresholds based on the absolute maximum of the underlying weight distribution which causes aggressive threshold changes during weight updates. Furthermore, the configurability of quantization levels of our approach allows n-ary weight representations. For instance, the quaternary representations requires the same amount of bits as ternary representations (thus, the same compression ratio) but achieves a significantly higher accuracy. Table 4 summarizes the accuracy differences of both

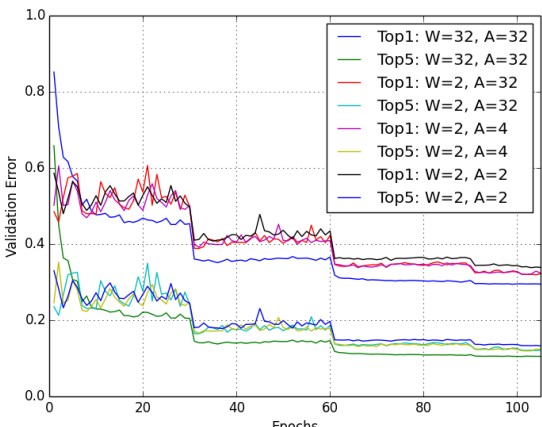

Figure 2: Validation error (Top1, Top5) of ResNet-18 on ImageNet for different bit-width combinations (W=2 refers to ternary weights).

Table 4: Validation accuracy (Top1, Top5) of ResNet18 and Inception-BN on ImageNet for 2-bit weights.

|  | Weights | Activations | Top1 | Top5 |
|---|---|---|---|---|
| ResNet-18 | bits | bits | % | % |
| TTQ | 2 | 32 | 67.0 | 87.3 |
| Ours (ternary) | 2 | 32 | 68.2 | 88.0 |
| Ours (quaternary-) | 2 | 32 | 69.2 | 88.7 |
| Inception-BN | bits | bits | % | % |
| TTQ | 2 | 32 | 67.0 | 87.3 |
| Ours (ternary) | 2 | 32 | 69.6 | 89.1 |
| Ours (quaternary-) | 2 | 32 | 70.8 | 89.8 |

approaches. All results are obtained using identical models, training time and hyperparameters.

**Impact of the nested clustering:** We validate the effectiveness of the nested-mean clustering by comparing it to quantile clustering. Assuming that the weights are approximately Gaussian distributed, we compute the cluster thresholds so that each cluster approximately contains a pre-specified amount of weights. Let $\Phi^{-1}(p)$ be the quantile function of the standard Gaussian distribution with zero mean and unit variance. Given a vector $\boldsymbol{p} = (p_1, \ldots, p_L)$ of pre-specified cluster probabilities that sum to one, the thresholds are computed as $\delta_i^l = \mu_w^l + \sigma_w^l \Phi^{-1}(\sum_{j \leq i} p_j)$ where $\mu_w^l$ and $\sigma_w^l$ are the mean and the standard deviation of the weights in layer $l$, respectively. Table 5

Table 5: Validation accuracy (Top1,Top5) using quinary weights of nested-means clustering and quantile-clustering for several quantiles for ResNet18 on ImageNet.

| Clustering Method | | | | | Top1 [%] | Top5 [%] |
|---|---|---|---|---|---|---|
| Nested-Mean | | | | | 69.7 | 89.0 |
| $p(\alpha_{-2})$ | $p(\alpha_{-1})$ | $p(0)$ | $p(\alpha_{+1})$ | $p(\alpha_{+2})$ | | |
| 20% | 20% | 20% | 20% | 20% | 68.8 | 88.5 |
| 11% | 22% | 33% | 22% | 11% | 69.3 | 88.8 |
| 8% | 17% | 50% | 17% | 8% | 69.4 | 88.9 |
| 6% | 11% | 66% | 11% | 6% | 69.2 | 88.8 |

summarizes the accuracy using quinary weights of both nested-means clustering and quantile clustering for several cluster sizes $\boldsymbol{p}$. We start with equal cluster sizes and incrementally increase the

cluster size of smaller weights while at the same time decreasing the cluster size of larger weights. The accuracy improves if we assign larger clusters to small weights and smaller clusters to large weights which validates our hypothesis. We want to emphasize that quantile-clustering assumes a Gaussian weight distribution which nested-means clustering does not.

## 6.4 RESOURCE EFFICIENCY

We summarize the parameter requirements and compression ratios, including the first and last layer of the ResNet18 model, in Table 6. The computational efficiency is shown on the example of a

Table 6: Parameter requirements of n-ary weight representations on ResNet18 on ImageNet.

| Weights | Parameter [MB] | Compression |
|---|---|---|
| Float32 | 46.7 | 1.0x |
| Quinary | 6.3 | 7.4x |
| Quaternary/Ternary | 4.9 | 9.5x |

typical ResNet layer in Table 7. As described in Sec. 3, we target reduce-and-scale inference which lowers most operations to reduced-precision additions and requires one full-precision multiplication per quantization level and per output feature. Please note, that we add another 7 bits to the bit width of activations for the addition operations in order to prevent overflows (we cannot rely on the normalization ability of floating-point arithmetic on integer hardware). Our approach removes

Table 7: Computational requirements of n-ary weight representations and fixed-point activations on a convolution layer with input-shape = (128 channels, 28 rows, 28 cols) and filter-shape = (128 filters, 128 channels, 3 rows, 3 cols).

| Weights | Activations | Sparsity | Multiplications | Workload | Additions | Workload |
|---|---|---|---|---|---|---|
| Float 32 | Float32 | 0% | $231.2 \cdot 10^6$ | 100.0% | $231.2 \cdot 10^6$ | 100.0% |
| Quinary | 8 bit | 53% | $401.4 \cdot 10^3$ | 0.2% | $108.7 \cdot 10^6$ | 22.0% |
| Quaternary | 8 bit | 57% | $301.1 \cdot 10^3$ | 0.1% | $99.4 \cdot 10^6$ | 20.2% |
| Ternary | 8 bit | 61% | $200.7 \cdot 10^3$ | 0.1% | $90.2 \cdot 10^6$ | 18.3% |
| Ternary | 4 bit | 61% | $200.7 \cdot 10^3$ | 0.1% | $90.2 \cdot 10^6$ | 13.4% |
| Ternary | 2 bit | 61% | $200.7 \cdot 10^3$ | 0.1% | $90.2 \cdot 10^6$ | 11.0% |

almost the complete multiplication workload due to the extremely low amount of quantization levels. The sparsity in the filter weights and the reduced bit-width of activations improves the addition workload by a factor of 4.5x to 9.1x.

## 7 CONCLUSION

We have presented a novel approach for compressing CNNs through quantization and connection pruning, which reduces the resolution of weights and activations and is scalable in terms of the number of quantization levels. As a result, the computational complexity and memory requirements of DNNs are substantially reduced, and an execution on resource-constrained devices is more feasible. We introduced a nested-means clustering algorithm for weight quantization that finds suitable interval thresholds that are subsequently used to assign each weight to a trainable scaling factor. Our approach exhibits both a low computational complexity and robustness to weight updates, which makes it an attractive alternative to other clustering methods. Furthermore, the proposed quantization method is flexible as it allows for various numbers of quantization levels, enabling high compression rates while achieving prediction accuracies close to single-precision floating-point weights. For instance, we utilize this flexibility to add an extra quantization level to ternary weights (quaternary weights), resulting in an improvement in prediction accuracy while keeping the bit width at two. For activation quantization, we developed an approximation based on statistical attributes that have been observed when batch normalization is employed. Experiments using state-of-the-art DNN

architectures on real-world tasks, including ResNet-18 and ImageNet, show the effectiveness of our approach.

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

## A   ACTIVATION QUANTIZATION

The truncated activations are uniformly quantized to k-bits:

$$Q_a(x) = \underbrace{\frac{\gamma}{2^k - 1}}_{\beta} \cdot \underbrace{\text{round}\left(\frac{2^k - 1}{\gamma} Q_c(x)\right)}_{\text{k-bit integer}},$$
(6)

where $\beta = \gamma/(2^k - 1)$ is a constant scaling factor. We approximate the gradient of the quantization function with the STE as

$$\frac{\partial E}{\partial x} = \frac{\partial E}{\partial Q_a}.$$
(7)

## B   COMPARISON TO RELATED WORK

We compare the validation accuracy of different bit-width combinations of our approach to the most related work in Table 8. We also report the amount of sparsity and increase in training time (if it was reported by the authors). The results are obtained from the original publications with the exception of TTQ on Inception-BN, which we reproduced ourselves. Our method outperforms previous work for each bit-width combination of weights and activations. Furthermore, the simplicity of computing the nested-means clustering and the linear transformation of the activations is less computationally intensive and, therefore, results in less training time than previously reported results.

---

[0]Full-precision short cut in the ResNet model.

Table 8: Comparison to related work (bold is ours) on the validation accuracy (Top1, Top5), increase in training time and sparsity of ResNet18 and Inception-BN on ImageNet.

| | Weights | Activations | Training | Top1 | Top5 | Sparsity |
|---|---|---|---|---|---|---|
| ResNet-18 | bits | bits | | % | % | % |
| Bi-Real Net (Liu et al., 2018b) | 1 | 1 | – | 56.4 | 79.5 | – |
| HWGQ (Cai et al., 2017) | 1 | 2 | – | 59.6 | 82.2 | – |
| SYQ (Faraone et al., 2018) | 1 | 8 | – | 62.9 | 84.6 | – |
| BWN (Rastegari et al., 2016) | 1 | 32 | – | 60.8 | 83.0 | – |
| PACT (Choi et al., 2018b) | 2 | 2 | – | 64.4 | – | – |
| PACT-SAWB (Choi et al., 2018a) | $2^0$ | $2^0$ | – | 67.0 | – | – |
| LQ-Net (Zhang et al., 2018) | 2 | 2 | 2.3x | 64.9 | 85.9 | – |
| Ours (ternary) | 2 | 2 | 1.2x | **66.2** | **86.7** | 61 |
| Ours (ternary) | 2 | 4 | 1.2x | **68.1** | **88.1** | 61 |
| SYQ (Faraone et al., 2018) | 2 | 8 | – | 67.7 | 87.8 | – |
| Ours (ternary) | 2 | 8 | 1.2x | **68.1** | **88.0** | 61 |
| TWN (Li & Liu, 2016) | 2 | 32 | – | 61.8 | 84.2 | – |
| INQ (Zhou et al., 2017) | 2 | 32 | – | 66.0 | 87.1 | – |
| TTQ (Zhu et al., 2016) | 2 | 32 | 1.0x | 66.6 | 87.2 | – |
| LQ-Net (Zhang et al., 2018) | 2 | 32 | 1.4x | 68.0 | 88.0 | – |
| Ours (ternary) | 2 | 32 | 1.2x | **68.2** | **88.0** | 61 |
| Ours (quaternary-) | 2 | 32 | 1.6x | **69.2** | **88.7** | 57 |
| ABC-Net (Lin et al., 2017) | 3 | 3 | – | 61.0 | 83.2 | – |
| PACT (Choi et al., 2018b) | 3 | 3 | – | 68.1 | – | – |
| LQ-Net (Zhang et al., 2018) | 3 | 3 | 3.7x | 68.2 | 87.9 | – |
| LQ-Net (Zhang et al., 2018) | 3 | 32 | 1.7x | 69.3 | 88.8 | – |
| INQ (Zhou et al., 2017) | 3 | 32 | – | 68.1 | 88.4 | – |
| Ours (quinary) | 3 | 32 | 2.0x | **69.7** | **89.0** | 53 |
| PACT (Choi et al., 2018b) | 4 | 4 | – | 69.2 | – | – |
| LQ-Net (Zhang et al., 2018) | 4 | 4 | – | 69.3 | 88.8 | – |
| UNIQ (Baskin et al., 2018) | 4 | 8 | – | 67.0 | – | – |
| INQ (Zhou et al., 2017) | 4 | 32 | – | 68.9 | 89.0 | – |
| LQ-Net (Zhang et al., 2018) | 4 | 32 | – | 70.0 | 89.1 | – |
| ABC-Net (Lin et al., 2017) | 5 | 5 | – | 65.0 | 85.9 | – |
| PACT (Choi et al., 2018b) | 5 | 5 | – | 69.8 | – | – |
| UNIQ (Baskin et al., 2018) | 5 | 8 | – | 68.0 | – | – |
| INQ (Zhou et al., 2017) | 5 | 32 | – | 69.0 | 89.1 | – |
| Pruning (Liu et al., 2018a) | 32 | 32 | – | 66.5 | 87.3 | 40 |
| Winograd-ReLU (Liu et al., 2018a) | 32 | 32 | – | 66.6 | 87.4 | 65 |
| Inception-BN | bits | bits | | % | % | % |
| TTQ | 2 | 32 | 1.0x | 67.0 | 87.3 | 58 |
| Ours (ternary) | 2 | 32 | 1.0x | **69.6** | **89.1** | 58 |
| Ours (quaternary-) | 2 | 32 | 1.1x | **70.8** | **89.8** | 52 |
| Ours (quinary) | 3 | 32 | 1.1x | **71.7** | **90.3** | 54 |

