# OpenReview forum: "N-Ary Quantization for CNN Model Compression and Inference Acceleration"
_ICLR.cc/2019/Conference_

### Official Review · AnonReviewer3 · 2018-11-04
**A good work in CNN model compression**

**Rating:** 7
**Confidence:** 5

**Review:**

This paper proposes to use n-ary representations for convolutional neural network model quantization. A novel strategy of nested-means clustering is developed to update weights. Batch normalization is also considered in the activation quantization. Experiments on both weight quantization and activation quantization are conducted and show effectiveness.

Strengths:
1.	The idea of nested-means clustering is interesting, which somehow shows its effectiveness.
2.	State-of-the-art experimental results.
3.	The representation is excellent, and it is easy to follow.

Concerns:
1.	Though the experiment study seems solid, an ablation study is still missing. For example, how important is the nested-means clustering technique? What is the effect if replacing it with the original one or with other clustering methods? What will happen if expanding the interval in the quantization of activation? All these kinds of questions are hard to answer without an ablation study.
2.	It is not clear how the weight and activation quantization are addressed together.
3.	If counting the first and last layers, what is the size of the model (the number of parameters)?
4.	Similarly, what are the FLOPs in different settings of experiments? This seems missing.
5.	When discussing the related work about model compression, there are important references missing. I just list two references in the latest vision and learning literature:
[Ref1] X. Lin et al. Towards accurate binary convolutional neural network. NIPS 2017
[Ref2] Z. Liu et al. Bi-Real Net: Enhancing the Performance of 1-bit CNNs with Improved Representational Capability and Advanced Training Algorithm. ECCV 2018.

---

> ### Author Response · Authors · 2018-11-26
> **Regarding your concerns**
>
> Many thanks for the valuable feedback, we addressed all concerns in the revised version of the paper. In particular:
>
> - Ablation study: we agree, an ablation study is required to show the actual benefits of the nested-means clustering. We added the study in the revised submission. However, we were not able to finish the study on the activation clipping interval but we would include it in the next revision as well.
>
> - Weight and activation quantization: both, weight and activation quantization are required for inference acceleration. We added more information on how to efficiently calculate these representations for the inference.
>
> - Model size and FLOPs: we include an evaluation of both in Sec. 6+7
>
> - Related work: selection was limited due to space constraints, but we now also included the provided references.

---

### Official Review · AnonReviewer2 · 2018-11-07
**in-depth analysis is needed for this paper**

**Rating:** 4
**Confidence:** 4

**Review:**

This paper is about CNN model compression and inference acceleration using quantization. The main idea is to use 'nest' clustering for weight quantization, more specifically, it partitions the weight values by recurring partitioning the weights by arithmetic means and negative of that of that weight clustering.

I have several questions for this paper:

1) the main algorithm is mainly based on the hypothesis that the weights are with Gaussian distribution. What happens if the weights are not Gaussian, such as skewed distribution? Seems the outliners will bring lots of issues for this nest clustering  for partitioning the weight values.

2) Since the paper is on inference acceleration, there is no real inference time result. I think having some real inference time on these quantized models and showing how their inference time speedup is will be interesting.

3) Activation quantization in Section 4 is a standard way for quantization, but I am curious how to filter out the outliner, and how to set the clipping interval?

4) I am not sure what does the 'sparsity' mean in Table 2? Does this quantization scheme introduce many zeros? Or sparsity is corresponding to the compression ratio? If that is the case, then many quantization algorithms can actually achieve better compression ratios with 2 bits quantization.

---

> ### Author Response · Authors · 2018-11-26
> **Regarding your questions**
>
> We appreciate your feedback on our initial submission. Regarding your questions:
>
> 1.  Gaussian distribution: We observe that l2-regularized weights are close to a zero-mean Gaussian, but actually we only assume that weights are symmetrically distributed around zero (the assumption is only used in nested-means clustering for the initial split at zero). That is, our clustering is also compatible with non-Gaussian distributions and we rephrased our text to be clearer in this regard. Nevertheless, the empirical observation that weights are close to Gaussian is seconded by other work [1][2].
>
> 2. The focus of this work is on the concept of this quantization scheme, but we admit that more details on reduction of memory footprint and computational workload would be helpful, which is now included in the revised submission (see Table 6 and 7). A detailed study on inference performance on multiple architectures is beyond the scope of this work, as in our experience such experiments require various code optimizations. Otherwise, there would be little value in reporting such performance numbers.
>
> 3. This is correct, the activation quantization is a standard way for transforming floating-point values into an integer format. Section 4 is discussing an appropriate clipping interval (including how to select the interval) that can filter out the outliers. Our experimental results indicate that this selection is appropriate.
>
> 4. Sparsity refers to the percentage of zero-valued elements in the weights. For instance, 60% sparsity means that 60% of the weights are zero.
>
> [1] Chaim Baskin et. al. UNIQ: uniform noise injection for the quantization of neural networks
> [2] Charles Blundell et. al. Weight Uncertainty in Neural Networks

---

### Official Review · AnonReviewer4 · 2018-11-11
**Limited novelty**

**Rating:** 4
**Confidence:** 4

**Review:**

Summary: This paper proposes a technique for quantizing the weights and activations of a CNN. The main contribution is in replacing the heuristic to find good quantization intervals of (Zhu et al, 2016) with a different heuristic based on a hierarchical clustering algorithm, and empirically validating its effectiveness.

Strenghts:
- The proposed nested-means heuristic is simple and makes sense intuitively.
- The experiments on two modern architectures seem solid and demonstrate good empirical performance.

Weaknesses:
- The main weakness is the limited novelty of this paper. The proposed setup is almost identical to the one in (Zhu et al, 2016), except for the replacement of the heuristic to find quantization intervals with another one. While the experiments demonstrate the empirical effectiveness of the method as a whole, what is missing is a direct, controlled comparison between the original heuristic and the proposed one. Now it is hard to tell whether the accuracy increases are obtained through the proposed adaptation or because of other factors such as a better implementation or longer training.
- In section 4, it is not made clear whether the activations are quantized according to the same scheme as the weights (apart from the issue of selecting a good clipping interval, which is addressed).
- The paper is a bit short on references, considering the many recent works on quantized neural networks.

Minor comments and questions:
- The wording is sometimes imprecise, making some arguments hard to follow. Two examples:
-- "Lowering the learning rate for re-training can diminish heavy changes in the weight distribution, at the cost of longer time to converge and the risk to get stuck at plateau regions, which is especially critical for trainable scaling factors"
-- "This approach is beneficial because it defines cluster thresholds which are influenced by large weights that were shown to play a more important role than smaller weights (Han et al., 2015b)"
- The title says "for compression and inference acceleration", so it would be nice if the paper reports some compression and timing metrics in the experiments section.
- The notation in section 3.1 overly complicated, could probably be simplified a bit for readability.
- Section 3.3: "However, having an additional hyperparameter t_i for each scaling factor alpha_i renders the mandatory hyperparameter tuning infeasible." -> From section 4.2 in (Zhu et al, 2016), I believe the constant factor t is shared across all layers, making it only a single hyperparameter.
- Last paragraph of section 4: "(Cai et al., 2017) experimentally showed that the pre-activation distribution after batch normalization are all close to a Gaussian with zero mean and unit variance. Therefore, we propose to select a fixed clipping parameter gamma.". -> But what about the activations *before* the batchnorm layer where the assumption of zero mean and unit variance does not hold?

---

> ### Author Response · Authors · 2018-11-26
> **Novelty, comparison and other comments**
>
> Many thanks for the detailed feedback, which helped us to (hopefully) improve the revised version of the paper
>
> - Novelty: the difference you point out to (Zhu et al, 2016) is correct, we adopted the gradient-based scaling factors. We evaluated several ways of obtaining the scaling factors but the approach of (Zhu et al, 2016) is the best performing. Our contributions are as follows: (1) a novel clustering approach that achieves better performance and allows for configurable quantization levels without additional hyperparameters. As a result, we achieve 2.6% better Top1 accuracy (Inception on ImageNet) for the ternary (2-bit encoding) representation. The configurable quantization levels enables the quaternary (2-bit encoding) representation which achieves 3.8% better Top1 accuracy without increasing quantization footprint. (2) Activation quantization by arguing about appropriate clipping intervals. (3) An analysis of the inference workload using reduce-scale architecture that minimizes the number of multiplications and substantially reduces the amount of additions.
>
> - Comparison to (Zhu et al, 2016): as part of our result discussion we compare to (Zhu et al, 2016), with the same training parameters as for our quantization. The accuracy we obtain is actually higher than the one reported in (Zhu et al, 2016), most likely because we used adaptive learning rate. We hope that this methodology demonstrates the improvement of this quantization compared to prior work.
>
> - Activation quantization: we quantize activations differently to weights by a simple linear transformation, because non-uniform activations are extremely difficult to implement efficiently for inference. We addressed this issue in the revised submission.
>
> - References: please refer to the general comments, where we discuss our limited selection due to space constraints.
>
> - Notation: we changed the notation to a less cluttered notation.
>
> - Hyperparameter t: this is correct, t is shared across all layers. However, the number of hyper parameters increases if multiple quantization levels are used.
>
> - Batchnormalization: the output of the batchnorm layer is the input for the next convolution layer. Hence, we don’t have to quantize activations *before* the batchnorm layer in order to accelerate convolutions.

---

### Public Comment · (anonymous) · 2018-10-01
**Some positive and negative comments.**

Pos:
1:  For weight quantization, the authors propose to use "NESTED-MEANS CLUSTERING" to learn non-uniform weights quantization.
2:  Experiment is solid and the performance is convincing.

Neg:
1:  Some important references are missing for activation quantization. These two papers propose to learn activation clip scales and have observed significant performance boost.
[1]: PACT: PARAMETERIZED CLIPPING ACTIVATION FOR QUANTIZED NEURAL NETWORKS.
[2]: Bridging the Accuracy Gap for 2-bit Quantized Neural Networks (QNN)

2: The quantization functions for weights and activations are somehow incremental.

3: In my experience, adding "0" into representation is extremely important to the final performance. But I did not find the results in Table 4 using "Quaternary" when comparing to other approaches. I think this is a little unfair.

4: Adding "0" into representation is a trade-off between accuracy and inference efficiency. Because you cannot merely employ XNOR operations in bit-wise operations. Specifically, you have to use AND and XNOR operations with judgement, which increases implementation difficulty on hard-ware platforms.

---

> ### Public Comment · (anonymous) · 2018-12-06
> **Missing references**
>
> To add to the list of missing references, this paper also does n-ary quantization but it does not use nested means.
> https://arxiv.org/abs/1811.04985

---

### Author Response · Authors · 2018-11-26
**General comments to the submission**

- We would like to clarify an important difference to previous work that we might not have expressed clearly before. While most recent related work on quantization focuses on binarization and related concepts, which are in particular based on uniform quantization and result in computations based on population count instructions (XNOR/AND and similar work [1][2]), our concept is based on non-uniform quantization (similar to the one proposed in [3][4]) and results in reduce-scale computations [5]. As a result, we can avoid the costly popcount and instead rely on many additions followed by one multiplication per quantisation level and per output feature. As additions are much cheaper than multiplications [6], this concept directly addresses inference acceleration.

- Our main results are: (1) we can substantially reduce model footprint and the computational workload (number of operations respectively their precision/type/bitwidth), (2) nested-means seems to be very suitable for neural networks quantization as it partitions in a way that large weights are accurately represented, (3) the resulting performance of such quantized models outperforms prior work, including LQ-Net and TTQ (which we compared by their reported performance and by our own training experiments). We believe these insights to be of value for the research community.

-References: There is more work than can be covered given the existing space constraints, so we faithfully selected the most important work (according to our opinion). We believe Table 8 to present a comprehensive overview, but would be happy to extend this as long as readability is maintained. Furthermore, we included the references provided by the reviewers. We believe LQ-Net to be the currently most advanced work, which we actually outperform in accuracy.

[1] Matthieu Courbariaux and Yoshua Bengio. Binarynet: Training deep neural networks with weights and activations constrained to +1 or -1.
[2] Shuchang Zhou, Zekun Ni, Xinyu Zhou, He Wen, Yuxin Wu, and Yuheng Zou. Dorefa-net: Training low bitwidth convolutional neural networks with low bitwidth gradients.
[3] Song Han, Huizi Mao, and William J. Dally. Deep compression: Compressing deep neural network with pruning, trained quantization and huffman coding.
[4] Incremental network quantization: Towards lossless cnns with low-precision weights.
[5] G. Schindler, M. Zöhrer, F. Pernkopf, and H. Fröning. Towards efficient forward propagation on resource-constrained systems.
[6] Mark Horowitz. 1.1 computing’s energy problem (and what we can do about it).

---

### Meta-Review · Area_Chair1 · 2018-12-16
**Area chair recommendation**

**Confidence:** 5
**Recommendation:** Reject

**Metareview:**

The submission proposes a hierarchical clustering approach (nested-means clustering) to determine good quantization intervals for non-uniform quantization.  An empirical validation shows improvement over a very closely related approach (Zhu et al, 2016).

There was an overall consensus that the literature review was insufficient in its initial form.  The authors have proposed to extend it somewhat.  Other concerns are related to the novelty of the technique (R4 was particularly concerned about novelty over Zhu et al, 2016).

Two reviewers were against acceptance, and one was positive about the paper.  Due to the overall concerns about the novelty of the approach, and that these concerns were confirmed in discussion after the rebuttal, this paper is unlikely to meet the threshold for acceptance to ICLR.